# The Antimicrobial Susceptibility of *Porphyromonas gingivalis*: Genetic Repertoire, Global Phenotype, and Review of the Literature

**DOI:** 10.3390/antibiotics10121438

**Published:** 2021-11-24

**Authors:** Georg Conrads, Tim Klomp, Dongmei Deng, Johannes-Simon Wenzler, Andreas Braun, Mohamed M. H. Abdelbary

**Affiliations:** 1Division of Oral Microbiology and Immunology, Department of Operative Dentistry, Periodontology and Preventive Dentistry, Rheinisch-Westfälische Technische Hochschule University Hospital, 52074 Aachen, Germany; tklomp@ukaachen.de (T.K.); mabdelbary@ukaachen.de (M.M.H.A.); 2Department of Operative Dentistry, Periodontology and Preventive Dentistry, Rheinisch-Westfälische Technische Hochschule University Hospital, 52074 Aachen, Germany; jwenzler@ukaachen.de (J.-S.W.); anbraun@ukaachen.de (A.B.); 3Department of Preventive Dentistry, Academic Centre for Dentistry Amsterdam (ACTA), University of Amsterdam and VU University Amsterdam, 1081 LA Amsterdam, The Netherlands; d.deng@acta.nl

**Keywords:** *Porphyromonas gingivalis*, antimicrobial susceptibility, breakpoints, periodontal diseases, minimal inhibitory concentration, antibiotic stewardship

## Abstract

The in vitro antimicrobial susceptibility of 29 strains of the major periodontal pathogen *Porphyromonas gingivalis* and three *P. gulae* (as an ancestor) to nine antibiotics (amoxicillin, amoxicillin/clavulanate, clindamycin, metronidazole, moxifloxacin, doxycycline, azithromycin, imipenem, and cefoxitin) was evaluated by E-testing of minimal inhibitory concentration (MIC) according to international standards. The results were compared with 16 international studies reporting MICs from 1993 until recently. In addition, 77 currently available *P. gingivalis* genomes were screened for antimicrobial resistance genes. E-testing revealed a 100% sensitivity of *P. gingivalis* and *P. gulae* to all antibiotics. This was independent of the isolation year (1970 until 2021) or region, including rural areas in Indonesia and Africa. Regarding studies worldwide (675 strains), several method varieties regarding medium, McFarland inoculation standards (0.5–2) and incubation time (48–168 h) were used for MIC-testing. Overall, no resistances have been reported for amoxicillin + clavulanate, cefoxitin, and imipenem. Few strains showed intermediate susceptibility or resistance to amoxicillin and metronidazole, with the latter needing both confirmation and attention. The only antibiotics which might fail in the treatment of *P. gingivalis*-associated mixed anaerobic infections are clindamycin, macrolides, and tetracyclines, corresponding to the resistance genes *erm*(*B*), *erm*(*F*), and *tet*(*Q*) detected in our study here, as well as fluoroquinolones. Periodical antibiotic susceptibility testing is necessary to determine the efficacy of antimicrobial agents and to optimize antibiotic stewardship.

## 1. Introduction

*Porphyromonas gingivalis* is a non-motile, Gram-negative, rod-shaped/filamentous (pleomorphic), anaerobic bacterium forming black-pigmented colonies after 3–4 days of incubation on supplemented (vitamin K, hemin) blood agar plates. It is an opportunistic pathogenic bacterium, commonly found in the human body and especially in the oral cavity, where it is associated with periodontal diseases. Together with *Tannerella forsythia* and *Treponema denticola*, it forms the red complex of etiologically important bacteria and is regarded as THE key periodontal pathogen [1,2]. The detection rate of *P. gingivalis* increases with age [3]. In the periodontal sulcus *P. gingivalis* induces an inflammatory response upregulating the nutrient-rich flow of sulcus fluid causing bacterial overgrowth/proliferation and subsequently dysbiosis. Simultaneously, it impairs the bactericidal innate host defenses by blocking Toll-Like-Receptor 4 (TLR4) recognition, assuring its survival [4]. In addition to periodontitis, it has been frequently found in oral specimens from necrotizing ulcerative gingivitis, infected root canals, peri-implant lesions, and acute apical abscesses [5,6]. Besides the oral cavity, it has been detected at various body sites, such as intra-abdominally [7], vaginal samples in cases of vaginosis [8], amniotic fluid [9], synovial specimens of rheumatoid arthritis and psoriatic arthritis [10], and together with some other periodontal organisms, in occluded arteries of the lower extremities of Buerger’s disease patients [11]. Recently investigated links between *P. gingivalis* and age-related macular degeneration [12], adverse pregnancy outcomes [13], Alzheimer’s disease [14,15], atherosclerotic disease [16], and cancer [17] need confirmation.

From the clinical perspective and following appropriate antibiotic stewardship, systemic antibiotics should only be used in well-selected patients and cases. In periodontology, their application should be restricted to specific conditions of severe progressive periodontitis defined by international recommendations [18,19]. Metronidazole, amoxicillin (plus/minus metronidazole or clavulanate), and clindamycin, as well as (less common) certain fluoroquinolones (mainly ciprofloxacin), tetracyclines (including doxycycline) and macrolides (erythromycin or azithromycin), are the antibiotics used and always in conjunction with mechanical debridement. If the debridement is neglected, antibiotics might not reach the bacteria in biofilm with the consequence of much higher minimal inhibitory concentration (MIC) [20]. Importantly, only for non-oral, severe mixed anaerobic infections, cefoxitin and imipenem are established treatment options and therefore included here.

The present study aimed to test the in vitro susceptibility of clinical *P. gingivalis* strains, isolated from periodontal pockets in adult patients with advanced forms of periodontitis in Germany and worldwide, to all antibiotics mentioned above and by standardized methods. We selected strains according to the following criteria: of local importance (Germany) or to fill gaps of regions under-represented in MIC testing so far (namely Indonesia, Kenya, Canada). We also included three *P. gulae* strains (isolated from Squirrel monkeys, *Saimiri squirrius*) as the probable ancestor of *P. gingivalis* [21] and known for their ability to acquire nitroimidazole resistance [22]. In addition to phenotypic testing, we retrieved the sequence reads of 77 publicly available *P. gingivalis* genomes and searched for resistance genes. By discussing our results with respect to 16 international studies addressing MIC data from 1993 until recently, we intended to meta-analyze the global antimicrobial susceptibility of *P. gingivalis* over time, as a snapshot from its origin until 2021.

## 2. Results

### 2.1. Antibacterial Susceptibility Pattern of Our Strain Collection

The MIC of our strain collection of 29 *P. gingivalis* and three *P. gulae* isolates to nine antibiotics are presented in Table 1. Next, we calculated the MIC range, MIC50 (median MIC value at which ≥50% of the isolates in a test population are inhibited) and MIC90 (90% of strains susceptible), and results are presented in Table 2. Our MIC range, MIC50, and MIC90 were all in line with other studies conducted worldwide. No single strain of our collection reached resistance according to the breakpoints, so far defined by CLSI/EUCAST. In a few cases (strains AC07 from Germany and 083-02 from Indonesia), individual colonies showed a reduced susceptibility and were re-tested, but reached only a slightly higher, non-resistance-indicative MIC. The length of incubation needed for a clear MIC reading differed between strains (48 h, as recommended, up to 72 h). We did not find any pattern of resistance development among our limited number of strains. Almost the same MICs were measured for *P. gulae* (proclaimed ancestor of *P. gingivalis*), for strains from rural regions, or for recently isolated strains of Western countries (e.g., Germany, Canada) with access to antibiotics. Comparing five pairs of isolates from the same Java-Indonesian patients in 1994 and 2002 (young population with a high prevalence of periodontal diseases [23]), no tendency for rising resistance was observed.

In general, *P. gingivalis* showed very different phenotypes in terms of time to pigmentation, encapsulation (apparent by slimy colonies), or colony size, partially explaining the difficulties in obtaining MIC readings.

### 2.2. Antibacterial Susceptibility Pattern Worldwide

After carrying out an extensive search for global data on the in vitro antimicrobial susceptibility of *P. gingivalis* (Figure 1), here, we review the results in comparison of region and time (Table 3). Interestingly, even though the methods for susceptibility testing underwent standardization processes, variations performed by several groups and in several regions can be recognized (Table 4, including references). Some of these variations might be due to the availability of certain agar media or E-testing strips. In addition, different lengths of incubation before MIC reading were reported which might be related to both, different agar/broth media used and cultivability (e.g., oxygen sensitivity) of strains included. Comparing MICs over all antibiotics tested, the smallest ranges of MICs (four dilution steps) were reported for cefoxitin (≤0.125–1 mg/L) and imipenem (≤0.016–0.12 mg/L), but both antibiotics were only tested in three and five studies, including ours, respectively, and always found to be effective. The only other antibiotic demonstrating 100% effectiveness against *P. gingivalis* was amoxicillin/clavulanate (MIC range <0.016–0.75 mg/L, with a breakpoint for resistance of ≥8 mg/L). For all other antibiotics tested, very few resistant strains were recognized worldwide as the principal reason for the range-extension.

### 2.3. Analysis of Antimicrobial Resistance Genes

The presence/absence of resistance genes was detected by various approaches that are described in the Materials and Methods section. In addition, we used two different databases, ARG-ANNOT [46] and CARD 2020 [47], that include 2223 and 2631 sequences of antimicrobial genes, respectively. Our analysis using the CARD database revealed that all 77 investigated assembled genomes carried the *pgpB* gene producing lipid A 4′-phosphatase, which is responsible for polymyxin B (a cationic polypeptide antibiotic) resistance and, as dependent on complex formation with similar peptides called LPS binding protein (LBP), for evading TLR4-sensing and killing of *P. gingivalis* [48,49]. Because of the nephro- and neurotoxicity, polymyxin B is only topically used (eye-, ear-, and wound-infections) and has no application in periodontology. Interestingly, only the two genomes ERX1066730 and ERX2022748 harbored any other genes encoding for antimicrobial resistance. These two genomes represented strains from Germany and the Netherlands (Appendix A), respectively. The Dutch ERX2022748 genome carried only the *erm*(*F*) gene. In contrast, the German ERX1066730 genome carried the following four resistance genes: *erm*(*B*) and *erm*(*F*) conferring resistance to macrolide, lincosamide, and streptogramin B (MLSB); the *cat*(*A1*) gene for the chloramphenicol-resistance (not of interest as no application in anaerobic infections); and the *tet*(*Q*) gene encoding for a ribosomal protection protein conferring resistance to tetracycline. These results were confirmed by an alternative pipeline and database. Resistance to fluoroquinolones in Gram-negative bacteria mostly occurs by two mechanisms: first, through mutations in the target enzyme DNA gyrase and topoisomerase IV and second, by reducing intracellular fluoroquinolones through efflux pumps. However, we did not detect any fluoroquinolone-resistance determinants among the 77 genomes investigated here. Resistance to metronidazole in Gram-negative anaerobes is known to be associated with nitroimidazole-resistance genes (*nimA–J* isoforms). These genes (located chromosomally or on plasmids) encode a 5-nitroimidazole reductase that converts nitro-imidazole to amino-imidazole, thus preventing the generation of bactericidal nitroso-residues [50]. Of note, we did not detect any *nim* isoforms here.

## 3. Discussion

*P. gingivalis* MIC range, MIC50, and MIC90 data measured in our very limited number of strains were all within the range found by other studies conducted worldwide. We did not observe any trend of increased antibiotic resistance comparing data from *P. gulae* (a possible ancestor of *P. gingivalis* as isolated from monkeys [21]), isolates from rural regions in Kenya or Indonesia, or isolates from the 1970s to recent ones. However, if integrating more strains and respecting outliers, the global situation has been different since the end of the 20th and the beginning of the 21st century. In Germany (1999), but especially in Colombia (2010, 2020 [pre-proof]), amoxicillin-resistant strains were isolated with MICs above the breakpoint of ≥8 mg/L [33,41,51,52]. In Colombia, resistance against clindamycin (breakpoints between ≥4 [EUCAST] and ≥8 mg/L [CLSI], 23.5% resistant strains according to authors) and metronidazole (breakpoints between ≥4 [EUCAST) and ≥32 mg/L [CLSI], 21.6% resistant strains in 2010, rising to 24.6% in 2020 [pre-proof] according to authors) were also reported with an MIC90 as high as ≥16 mg/mL for both antibiotics. Resistance to metronidazole of a single (1 out of 10 tested) *P. gingivalis* strain was reported from Pakistan in 2020 also [53]. Although *nim*-associated metronidazole resistance is not highly numerous globally, there are reports with a prevalence between <1% (*Prevotella*) and 4% (*Bacteroides*) from quite a few regions such as Pakistan, North India, and Kuwait, but also the USA and Europe [50,54]. As a matter of concern, a few metronidazole-resistant *P. gulae* strains with MICs > 512 µg/mL were already isolated from dogs [22]. Developing of resistance in pets is plausible as they are frequently treated for parasites, and the imidazole-derivative fenbendazole is a popular choice [55]. After in vitro metronidazole challenge with sub-inhibitory concentrations, an adaptation of *P. gingivalis* was also demonstrated [56]. On the other hand, resistance to metronidazole in humans, even after selection pressure by treatment [57], seems to be unlikely, which might be due to a fitness cost associated with the acquisition of *nim* genes. Of note, *nim* genes were never found here according to the limited literature addressing this topic [58], our genome analysis results, and the best of our knowledge.

Again from Columbia but from a different group, an intermediate susceptibility to tetracycline (reaching the CLSI-breakpoint of 8 mg/L) was reported applying M.I.C.Evaluator strips [45]. As two independent Colombian groups were reporting increased resistance by applying good standard testing procedures, these results seem to be plausible and a matter of concern, as germs of all kinds easily cross borders, very apparent with the current SARS-CoV-2 pandemic. Further reports of tetracycline-resistant strains came from Germany [33], and the identification of *tet*(*Q*) genes underlines this risk (in our study and [59]).

Finally, macrolides, even if not very much used for treating periodontitis or anaerobic infections, are interesting as used for prophylaxis of infective endocarditis perioperatively, including periodontal open flap operations. Here, breakpoints for anaerobes are only defined for erythromycin and by CLSI (≥2 mg/L), but MICs as high as 8 mg/L for erythromycin (found in Italy 2007, [39]) or between 16 mg/L (found in Brazil 2006, [38]) and 24 mg/L (found in Colombia recently as MIC90, [52] pre-proof status) for azithromycin may indicate intermediate susceptibility or resistance. To avoid over-interpreting the clinical impact of very few resistant strains, it might be more constructive to use MIC50 and MIC90 data provided for all studies, including ours, in Table 3. The average *P. gingivalis*-MIC50 over all studies and about 700 strains (by exclusion of the Colombia strains which explains differences to Table 3) are promising and were the following (in mg/L): amoxicillin <0.016–<0.25, AMC < 0.016–<0.125, clindamycin < 0.016–≤0.125, metronidazole <0.016, fluoroquinolones 0.06–0.5, tetracyclines 0.015–0.75, macrolides < 0.016–0.25, imipenem 0.015–≤0.125, and cefoxitin 0.06–≤0.125. The same is true for MIC90 data (again by exclusion of the top resistant Colombia strains): amoxicillin <0.016–0.064, AMC < 0.016–0.125, clindamycin < 0.016–≤0.125, metronidazole < 0.016–0.75, fluoroquinolones 0.032–2, tetracyclines 0.023–16, macrolides <0.016–2, imipenem ≤ 0.125–0.06, and cefoxitin ≤ 0.125–0.5). Taken together, the only antibiotics which seem to have lost activity since the very late 20th century are tetracyclines, macrolides (both in accordance with the resistance genes found in 77 genomes investigated in depth here) as well as fluoroquinolones, the latter inactivated by mutation of target enzyme or by efflux pump-based extrusion.

Because of a reason given below, some important studies were not included in our analysis in the first instance. However, the results of these six studies will be discussed below for further comparison. Jepsen et al. tested antibiotic susceptibility of 5323 *P. gingivalis* isolates from 2008–2015 by agar diffusion only [60]. The average non-susceptibility in 2008–2011 versus 2012–2015 for doxycycline, azithromycin, and ciprofloxacin was: 0% versus 0.04%, 0.64% versus 1.56%, and 4.55% versus 11.51%, respectively. Furthermore, the authors found a significant increase in clindamycin non-susceptibility of *P. gingivalis* (0.46% versus 1.72%, in this particular case confirmed by E-testing), which is not seen worldwide and might be due to its widespread use in German dental practices (29.3% of all antibiotic prescriptions) [61]. All 5323 strains tested were metronidazole- and amoxicillin/clavulanate-susceptible, and 99.62% were amoxicillin-susceptible. Recently, Kulik et al. investigated susceptibility patterns of 56 *P. gingivalis* strains among Swiss periodontitis patients from different decades [62]. Because of the applied MICRONAUT-S anaerobe MIC plates method, with microdilution only for confirmation of elevated MICs, this study was not included in the first instance but is important for the discussion. In summary, their strains yielded low MIC50 (0.0625–0.5 mg/L) and MIC90 (0.125–2 mg/L) values for all the antimicrobials tested with two isolates needing attention: one was *ermF*-positive and had MIC values higher than 8 mg/L, 2 mg/L, and 0.25 mg/L for clindamycin, azithromycin, and moxifloxacin, respectively. The second isolate had a high MIC value of 4 mg/L for moxifloxacin. Sequence analysis of the quinolone resistance-determining region (QRDR) of the *gyrA* gene confirmed a gene mutation, namely Ser-83 ≥ Phe substitution. Dahlen et al. revealed antibiotic susceptibility against seven antibiotics among 67 consecutive fresh isolates of *P. gingivalis* in a Swedish population, with data given only in figures but not as tables [63]. The strains showed an overall susceptibility to all tested antibiotics except for kanamycin. However, reduced sensitivity was detected in one strain for penicillin G (MIC 1 mg/L), in four strains for ampicillin (MIC > 0.5 mg/L), and in nine strains for clindamycin (MIC > 0.1 mg/L). In a Japanese study in 2007, 48 *P. gingivalis*/*P. endodontalis* strains were examined but results not sorted by species [64]. However, all of the 13 antibiotics tested were highly active against both species with only one strain found resistant to amoxicillin. The overall susceptibility of strains (27, with 25 of them *P. gingivalis*) in Japan was recently confirmed, but with 4.9% moxifloxacin and 22.8% clindamycin resistance as exceptions and high MIC90 values (64–128 mg/L) for macrolides [65].

Striking for many studies is the delay between the isolation of strains and publication of susceptibility data, which can be more than 5 years. Assuming that the antibiotic resistance would increase by the same rate as measured by Jepsen et al. [60], presently (2021) the number of non-susceptible strains for doxycycline, azithromycin, and ciprofloxacin could have reached 0.1%, 2.5%, and 19%, respectively. However, this is highly speculative, as the counteracting antibiotic stewardship might break this tendency.

Of final note, during this study, we came across susceptibility data of other *Porphyromonas* species also, and it must be concluded that *P. asaccharolytica, P. levii*, and *P. uenonis* were less susceptible to antibiotics [35,66]. For instance, two *P. asaccharolytica* and two *P. levii* strains from California were resistant to clindamycin (>32 mg/L). There were seven isolates (most likely non-*P. gingivalis*) with levofloxacin MICs of 4 mg/L and three with MICs of 8 mg/L [35]. Metronidazole resistance of non-oral *Porphyromonas* sp. with MIC < 256 mg/L was reported from Greece [67].

## 4. Materials and Methods

### 4.1. Bacterial Strains and Antibacterial Susceptibility Testing

The strains tested in our laboratory and their origin are summarized in Table 1 together with MIC values. Most historical strains are from the ACTA collection, and co-author D. Deng provided them with agreement of her institution. By means of a Google Scholar search, we could identify the most plausible country and year of strain isolation. The MICs of 29 *P. gingivalis* and 3 *P. gulae* isolates to nine antibiotics, including amoxicillin, amoxicillin/clavulanate, clindamycin, metronidazole, moxifloxacin (fluoroquinolone), doxycycline (tetracycline), and azithromycin (macrolide), as well as imipenem and cefoxitin (the latter two for severe infections only) were determined by E test method (AB Biodisk, Solna, Sweden). Bacterial strains were grown on Brucella blood agar plates (Becton Dickinson GmbH, Heidelberg, Germany), supplemented with 5% sheep blood, hemin (5 mg/L), and vitamin K1 (10 mg/L), for up to 5 days. The test strains (a few fresh colonies) were suspended in sterile phosphate-buffered saline equivalent to a 1.0 McFarland standard and streaked confluently over the surface blood agar plates with the aid of a sterile swab. Plates were incubated anaerobically for 2 to 5 days. Inhibition zones were measured at 48 h according to the recommendations of the manufacturer and the Clinical and Laboratory Standards Institute (CLSI), but for slow-growing strains, a longer incubation time was necessary before reading was possible. Percentages of resistant isolates were calculated using breakpoints advised by the CLSI (document M100-ED31:2021 Performance Standards for Antimicrobial Susceptibility Testing, 31st Edition with breakpoints publicly made available in Table 2J *MIC Breakpoints for Anaerobes*) and by the European Committee on Antimicrobial Susceptibility Testing (EUCAST, Clinical Breakpoint Tables v. 11.0, valid from 1 January 2021, at https://eucast.org/clinical_breakpoints/, accessed on 19 November 2021). For comparison with the literature, an advanced PubMed and Google Scholar search was performed combining MESH terms (antibiotic[MeSH Terms] AND inhibitory concentration, minimum[MeSH Terms] AND (*Porphyromonas gingivalis* [MeSH Terms] OR *Porphyromonas* [All fields])), with results illustrated in Figure 1.

### 4.2. Analysis of Antimicrobial Resistance Genes

In addition to phenotypic testing, we retrieved the sequence reads of 77 publicly available *P. gingivalis* genomes that are listed in Appendix A using *Bactopia* pipeline version 1.6.5 [68]. Within the Bactopia workflow, the quality check on the sequence reads was assigned, and reads below the quality requirements were filtered out. The remaining high-quality sequence reads were assembled using the *Shovill* pipeline version 1.1.0 (https://github.com/tseemann/shovill, accessed on 10 October 2021) and the default setting for the *SKESA* assembler version 2.4.0 [69]. Bactopia searches for antimicrobial resistance genes directly using the *ARIBA* pipeline version 2.14.6 [70] and the comprehensive antibiotic resistance database (*CARD* 2020) [47] as the default settings for predicting antibiotic resistance. In addition, we doubled checked for the presence or absence of acquired resistance genes by applying the *ABRicate* pipeline (version 0.8.132, https://github.com/tseemann/abricate, accessed on 10 October 2021) on the assembled genomes using default settings, and here, both the ARG-ANNOT (version V6, https://ifr48.timone.univ-mrs.fr/blast/arg-annot_v6.html, accessed on 10 October 2021) [46] and CARD 2020 (version 3.1.4, https://card.mcmaster.ca/, accessed on 10 October 2021) [47] public databases were used as references for detecting a wide variety of point mutations and reference sequences known to be associated with antimicrobial resistance. A list of all updated versions of the ARG-ANNOT database can be found under the following link: https://www.mediterranee-infection.com/acces-ressources/base-de-donnees/arg-annot-2/ (accessed on 10 October 2021).

## 5. Conclusions

Fortunately, antimicrobial resistance of *P. gingivalis* is not yet emerging but an increase of MIC data of tetracyclines, macrolides, lincosamide, and fluoroquinolones, has been recognized since the end of the 20th century. Only very few relevant genes, such as *erm*(*B*) or *erm*(*F*) conferring MLSB-resistance and *tet*(*Q*) encoding resistance to tetracyclines, were detected in the 77 publicly available genomes. Of note, neither phenotypic metronidazole resistance (with very few exceptions needing confirmation) nor corresponding *nim* genes were reported for *P. gingivalis*. However, a resistance-transfer from related *Bacteroides*, *Prevotella*, or other *Porphyromonas* species (including pet isolates) could emerge, as these are increasing in many regions worldwide. Thus, adjunctive antimicrobial usage in the treatment of periodontitis must be restricted and antibiotic stewardship and resistance gene screening extended.

## Figures and Tables

**Figure 1 antibiotics-10-01438-f001:**
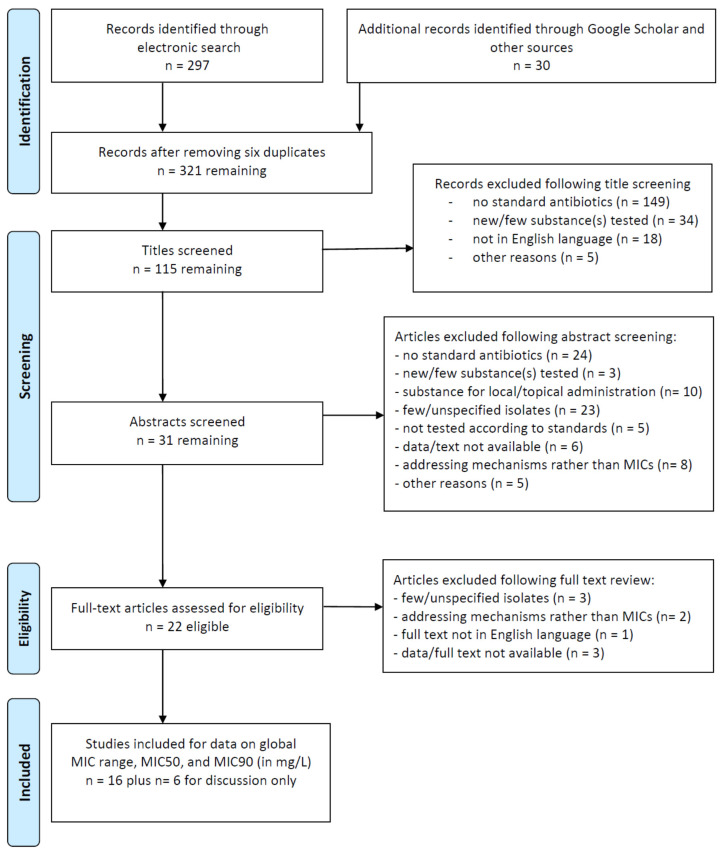
Flow diagram of the search for *Porphyromonas gingivalis*-/minimal inhibitory concentration (MIC)-related articles (based on the PRISMA checklist).

**Table 1 antibiotics-10-01438-t001:** In vitro antimicrobial susceptibility of 3 *Porphyromonas gulae* and 29 *P. gingivalis* strains (isolated between 1970 and 2021) to nine antibiotics determined by the E test (results in mg/L). Among the tested 31 strains, 20 were retrieved from previous studies while 11 strains were isolated and investigated for the first time during this study. The MIC range, together with MIC50 and MIC90 was calculated and is reported together with breakpoints in Table 2.

Strain	Country-Isolation Year	Reference	Amoxicillin	AMC	Clindamycin	Metronidazole	Moxifloxacin	Doxycycline	Azithromycin	Imipenem	Cefoxitin
*P. gulae* I-372	USA-FL-1986	Clark 1988 [24]	<0.016	<0.016	<0.016	<0.016	0.023	0.047	1	0.006	0.047
*P. gulae* I-433	USA-FL-1986	Clark 1988 [24]	<0.016	<0.016	<0.016	<0.016	0.003	0.016	0.094	0.012	0.064
*P. gulae* G251	USA-FL-1986	Clark 1988 [24]	<0.016	<0.016	<0.016	<0.016	0.012	0.047	0.19	0.016	0.19
W83	Germany-70th.	ATCC Coykendall 1980	<0.016	<0.016	<0.016	<0.016	0.002	0.016	n.d.	0.004	0.032
AJW5 = VAG 5	USA-NY-late 80th.	Lee 1991 [25]	<0.016	<0.016	<0.016	<0.016	<0.002	<0.016	0.19	0.006	<0.016
22KN6-12	Japan-late 80th.	Nagata 1991 [26]	<0.016	<0.016	<0.016	<0.016	0.004	<0.016	0.094	0.032	0.125
OMG 406	Kenya-Mid 80th.	Dahlen 1989 [27]	<0.016	<0.016	<0.016	<0.016	0.004	<0.016	0.094	0.004	0.023
RB22D-1 = ATCC 49417	Canada-early 90th.	Ménard 1995 [28]	<0.016	<0.016	<0.016	<0.016	0.012	0.032	0.38	0.008	0.044
7B5	Canada-early 90th.	Ménard 1995 [28]	<0.016	<0.016	<0.016	<0.016	0.008	0.047	0.19	0.023	0.032
23A4	Canada-early 90th.	Ménard 1995 [28]	<0.016	<0.016	<0.016	<0.016	0.012	0.047	0.25	0.008	0.19
HW24D-2	Canada-early 90th.	Ménard 1995 [28]	<0.016	<0.016	<0.016	<0.016	0.016	0.032	0.19	0.008	0.032
Indo021-94	Indonesia-1994	Timmerman 1998 [29]	<0.016	<0.016	<0.016	<0.016	<0.002	<0.016	0.19	0.003	<0.016
Indo021-02	Indonesia-2002	v.d. Velden 2006 [23]	<0.016	<0.016	<0.016	<0.016	<0.002	<0.016	0.125	0.006	<0.016
Indo059-94	Indonesia-1994	Timmerman 1998 [29]	<0.016	<0.016	<0.016	<0.016	0.004	<0.016	0.094	0.004	0.016
Indo059-02	Indonesia-2002	v.d. Velden 2006 [23]	<0.016	<0.016	<0.016	<0.016	0.012	0.064	0.19	0.016	0.125
Indo083-94	Indonesia-1994	Timmerman 1998 [29]	<0.016	<0.016	<0.016	<0.016	0.012	0.016	0.19	0.016	0.19
Indo083-02	Indonesia-2002	v.d. Velden 2006 [23]	<0.016	<0.016	<0.016	<0.016	0.016	0.023	0.19	0.094	0.064
Indo168-94	Indonesia-1994	Timmerman 1998 [29]	<0.016	<0.016	<0.016	<0.016	0.003	0.016	1.5	<0.002	0.047
Indo168-02	Indonesia-2002	v.d. Velden 2006 [23]	<0.016	<0.016	<0.016	<0.016	<0.002	<0.016	0.38	0.012	0.023
Indo210-94	Indonesia-1994	Timmerman 1998 [29]	<0.016	<0.016	<0.016	<0.016	0.016	0.064	0.38	0.064	0.38
Indo210-02	Indonesia-2002	v.d. Velden 2006 [23]	<0.016	<0.016	<0.016	<0.016	0.016	0.032	0.5	0.032	0.047
AC01	Germany-2021	this study 2021	<0.016	<0.016	<0.016	<0.016	<0.003	<0.016	0.38	0.008	<0.016
AC04	Germany-2021	this study 2021	0.016	0.016	<0.016	0.016	<0.012	0.032	1	0.023	0.094
AC07	Germany-2021	this study 2021	<0.016	<0.016	<0.016	<0.016	0.25	<0.016	0.19	0.002	<0.016
AC08	Germany-2021	this study 2021	<0.016	<0.016	<0.016	<0.016	<0.004	<0.016	0.064	0.012	0.19
AC26	Germany-2021	this study 2021	<0.016	<0.016	<0.016	<0.016	<0.006	<0.016	1.5	0.008	0.125
AC27	Germany-2021	this study 2021	<0.016	<0.016	<0.016	<0.016	<0.002	<0.016	0.5	0.004	<0.016
AC29	Germany-2021	this study 2021	<0.016	<0.016	<0.016	<0.016	0.003	<0.016	0.047	0.012	0.25
AC38	Germany-2021	this study 2021	<0.016	<0.016	<0.016	<0.016	0.002	<0.016	0.19	0.016	0.047
AC50	Germany-2021	this study 2021	<0.016	<0.016	<0.016	<0.016	0.003	<0.016	0.016	0.003	<0.016
AC58	Germany-2021	this study 2021	<0.016	<0.016	<0.016	<0.016	0.002	<0.016	0.094	0.023	0.032
AC71	Germany-2021	this study 2021	<0.016	<0.016	<0.016	<0.016	<0.002	<0.016	<0.016	0.002	0.016

AMC amoxicillin/clavulanate; the isolation time was roughly calculated or estimated if not explicitly mentioned in the text of publication.

**Table 2 antibiotics-10-01438-t002:** MIC range, together with MIC50 and MIC90, of 3 *Porphyromonas gulae* and 29 *P. gingivalis* strains subjected to this study. If defined, CLSI and EUCAST breakpoints are also given and the corresponding susceptibility.

		MIC (mg/L)			CLSI		EUCAST		Susceptibility (%)
Antibiotic	Range	50%	90%	S≤	I	R≥	S≤	R≥	
**Amoxicillin**	<0.016–0.016	<0.016	<0.016	0.5	1	2	0.5	2	100%
**AMC**	<0.016–0.016	<0.016	<0.016	4/2	8/4	16/8	4	8	100%
**Clindamycin**	<0.016	<0.016	<0.016	2	4	8	4	4	100%
**Metronidazole**	<0.016–0.016	<0.016	<0.016	8	16	32	4	4	100%
**Moxifloxacin**	<0.002–0.25	0.003	0.016	2	4	8	IE	IE	100%
**Doxycyclin**	<0.016–0.064	<0.016	0.047	4	8	16	E	E	100%
**Azithromycin**	<0.016–1.5	0.125	0.25	n.d.	n.d.	n.d.	n.d.	n.d.	100% **
**Imipenem**	<0.002–0.094	0.008	0.023	4	8	16	2	4	100%
**Cefoxitin**	<0.016–0.38	0.032	0.19	16	32	64	IE	IE	100%

S—susceptible, I—intermediate, R—resistant, n.d.—not defined; AMC—amoxicillin/clavulanate (breakpoint concentrations given for both substances). IE—insufficient evidence; E—evidence, but MIC-clinical outcome difficult to correlate; ** assumed, deduced from CLSI breakpoint for erythromycin S/I/R: 2/2/2.

**Table 3 antibiotics-10-01438-t003:** Comparison of 16 studies (1993–2019) determining the MIC range, MIC50, and MIC90 (in mg/L) of the key periodontopathogen *Porphyromonas gingivalis* for 9 antibiotics (including three classes) applying different methods (see Table 2).

Region [Ref.]	N Strains	Year Publication	MIC	Amoxicillin	AMC	Clindamycin	Metronidazole	Fluorochinolone	Tetracycline	Macrolide	Imipenem	Cefoxitin
Finland [30]	64	1993	Range	<0.016–0.023 *	n.d.	<0.016	<0.002–0.023	0.019–0.75 (Cip)	<0.016–0.047 (Dox)	≤0.016–0.19 (Ery)	n.d.	n.d.
Japan [31]	10	1995	Range	n.d.	n.d.	≤0.031	≤0.031–1	≤0.031–2 (Spa)	0.063–0.5 (Tet)	≤0.031–0.5 (Ery)	n.d.	n.d.
Spain [32]	31	1998	Range	≤0.125–1	n.d.	≤0.125	0.125–2	n.d.	≤0.125–0.5 (Tet)	≤0.125–1 (Ery)	≤0.125	≤0.125–0.25
Germany [33]	26	1999a	Range	≤0.25–16	n.d.	≤0.125–1	≤0.25–0.5	0.25–1 (Cip)	0.25–32 (Dox)	n.d.	n.d.	n.d.
Germany [34]	32	1999b	Range	n.d.	0.016–0.125	n.d.	0.002–0.5	n.d.	0.016–2 (Tet)	n.d.	n.d.	n.d.
International [35]	31	2004	Range	0.03–16 **	≤0.06–0.5	≤0.016–0.125	0.06–0.5	0.125–8 (Lev)	n.d.	n.d.	≤0.016–0.03	n.d.
Netherlands [36]	26	2005a	Range	<0.016	<0.016	<0.016	<0.016	0.001–2 (Cip)	0.015–0.32 (Tet)	0.015–1.5 (Azi)	n.d.	n.d.
Spain [36]	15	2005b	Range	<0.016	<0.016	<0.016	<0.016	0.15–0.75 (Cip)	0.25–1 (Tet)	<0.016 (Azi)	n.d.	n.d.
Turkey [37]	15	2005c	Range	n.d.	n.d.	0.03–0.12	0.06–0.5	n.d.	n.d.	n.d.	0.015–0.03	n.d.
Brazil [38]	20	2006	Range	0.016–1	0.016–0.125	0.016–0.125	0.016–1.5	n.d.	0.016–2 (Tet)	0.016–12 (Azi)	n.d.	n.d.
Italy [39]	32	2007	Range	n.d.	≤0.03–0.06	≤0.03–4	0.06–2	0.06–4 (Lev)	n.d.	≤0.03–8 (Ery)	≤0.03–0.12	0.06–1
Switzerland [40]	152	2008	Range	n.d.	<0.016–0.064	<0.016–0.125	<0.016–0.016	n.d.	<0.016–2 (Tet)	n.d.	n.d.	n.d.
Colombia [41]	51	2010	Range	0.016–>256	<0.016–0.064	0.08 to ≥16	0.08 to ≥16	0.006–0.032 (Mox)	<0.015–8 (Tet) ***	n.d.	n.d.	n.d.
Iran [42]	50	2011	Range	0.016–2	0.016–0.125	0.016–0.08	0.016->1	0.002–1 (Cip)	0.016–0.5 (Dox)	0.002–0.38 (Azi)	n.d.	n.d.
Netherlands [43]	50	2012	Range	<0.016–0.38	<0.016–0.25	<0.016	<0.016–0.032	n.d.	<0.016–0.75 (Tet)	<0.016–2 (Azi)	n.d.	n.d.
Morocco [44]	70	2019	Range	<0.016–0.75	<0.016–0.75	n.d.	<0.016–0.094	n.d.	n.d.	<0.016–1.5 (Azi)	n.d.	n.d.
MIC range over all studies	675	1993–2019	total MIC range	<0.016–>256	<0.016–0.75	<0.016 to ≥16	<0.002 to ≥16	0.001–8 ^$^	<0.016–32 ^$^	≤0.016–12 ^$^	≤0.016–0.12	≤0.125–1
MIC range this study	32	2021	MIC range	<0.016–0.016	<0.016–0.016	<0.016	<0.016–0.016	<0.002–0.25 (Mox)	<0.016–0.064 (Dox)	<0.016–1.5 (Azi)	<0.002–0.094	<0.016–0.38
Finland [30]	64	1993	MIC50	n.d.	n.d.	n.d.	n.d.	n.d.	n.d.	n.d.	n.d.	n.d.
Japan [31]	10	1995	MIC50	n.d.	n.d.	≤0.031	0.5	0.25	0.25	0.25	n.d.	n.d.
Spain [32]	31	1998	MIC50	≤0.125	n.d.	≤0.125	0.125	n.d.	0.25	0.25	≤0.125	≤0.125
Germany [33]	26	1999a	MIC50	≤0.25	n.d.	≤0.125	≤0.25	0.5	≤0.25	n.d.	n.d.	n.d.
Germany [34]	32	1999b	MIC50	n.d.	n.d.	n.d.	n.d.	n.d.	n.d.	n.d.	n.d.	n.d.
International [35]	31	2004	MIC50	≤0.125	≤0.125	≤0.016	≤0.125	0.5	n.d.	n.d.	≤0.016	n.d.
Netherlands [36]	26	2005a	MIC50	<0.016	<0.016	<0.016	<0.016	0.125	0.015	0.25	n.d.	n.d.
Spain [36]	15	2005b	MIC50	<0.016	<0.016	<0.016	<0.016	0.25	0.5	<0.016	n.d.	n.d.
Turkey [37]	15	2005c	MIC50	n.d.	n.d.	0.06	0.12	n.d.	n.d.	n.d.	0.015	n.d.
Brazil [38]	20	2006	MIC50	0.016	0.016	0.016	0.125	n.d.	0.032	0.25	n.d.	n.d.
Italy [39]	32	2007	MIC50	n.d.	0.06	≤0.03	0.06	0.06	n.d.	0.06	≤0.03	0.06
Switzerland [40]	152	2008	MIC50	n.d.	<0.016	<0.016	<0.016	n.d.	0.023	n.d.	n.d.	n.d.
Colombia [41]	51	2010	MIC50	0.125	<0.016	8	0.256	0.023	n.d.	n.d.	n.d.	n.d.
Iran [42]	50	2011	MIC50	0.024	0.016	0.016	0.016	0.094	0.032	0.032	n.d.	n.d.
Netherlands [43]	50	2012	MIC50	<0.016	<0.016	<0.016	<0.016	n.d.	0.023	<0.016	n.d.	n.d.
Morocco [44]	70	2019	MIC50	<0.016	<0.016	n.d.	<0.016	n.d.	n.d.	0.19	n.d.	n.d.
MIC50 range over all studies	675	1993–2019	MIC50 range	<0.016–<0.25	<0.016-<0.125	<0.016–8	<0.016	0.06–0.5	0.015–0.75	<0.016–0.25	0.015–≤0.125	0.06–≤0.125
MIC50 this study	32	2021	MIC 50	<0.016	<0.016	<0.016	<0.016	0.003 (Mox)	<0.016 (Dox)	0.125 (Azi)	0.008	0.032
Finland [30]	64	1993	MIC90	n.d.	n.d.	n.d.	n.d.	n.d.	n.d.	n.d.	n.d.	n.d.
Japan [31]	10	1995	MIC90	n.d.	n.d.	≤0.031	1	0.5	0.5	0.5	n.d.	n.d.
Spain [32]	31	1998	MIC90	0.25	n.d.	≤0.125	0.125	n.d.	1	1	≤0.125	≤0.125
Germany [33]	26	1999a	MIC90	≤0.25	n.d.	1	0.5	1	16	n.d.	n.d.	n.d.
Germany [34]	32	1999b	MIC90	n.d.	n.d.	n.d.	n.d.	n.d.	n.d.	n.d.	n.d.	n.d.
International [35]	31	2004	MIC90	0.25	≤0.125	0.06	0.5	2	n.d.	n.d.	0.03	n.d.
Netherlands [36]	26	2005a	MIC90	<0.016	<0.016	<0.016	<0.016	0.38	0.023	0.5	n.d.	n.d.
Spain [36]	15	2005b	MIC90	<0.016	<0.016	<0.016	<0.016	0.75	0.75	<0.016	n.d.	n.d.
Turkey [37]	15	2005c	MIC90	n.d.	n.d.	0.06	0.5	n.d.	n.d.	n.d.	0.015	n.d.
Brazil [38]	20	2006	MIC90	0.125	0.064	0.047	0.75	n.d.	0.75	2	n.d.	n.d.
Italy [39]	32	2007	MIC90	n.d.	0.06	0.06	1	0.12	n.d.	0.5	0.06	0.5
Switzerland [40]	152	2008	MIC90	n.d.	<0.016	<0.016	<0.016	n.d.	0.19	n.d.	n.d.	n.d.
Colombia [41]	51	2010	MIC90	>256	<0.016	≥16	≥16	0.032	n.d.	n.d.	n.d.	n.d.
Iran [42]	50	2011	MIC90	1	0.125	0.047	0.5	0.75	0.5	0.38	n.d.	n.d.
Netherlands [43]	50	2012	MIC90	<0.016	<0.016	<0.016	<0.016	n.d.	0.25	0.094	n.d.	n.d.
Morocco [44]	70	2019	MIC90	0.064	0.032	n.d.	0.047	n.d.	n.d.	1	n.d.	n.d.
MIC90 range over all studies	675	1993–2019	MIC90 range	<0.016–>256	<0.016–0.125	<0.016–≥16	<0.016–≥16	0.032–2	0.023–16	<0.016–2	≤0.125–0.06	≤0.125–0.5
This study	32	2021	MIC 90	<0.016	<0.016	<0.016	<0.016	0.016 (Mox)	0.047 (Dox)	0.25 (Azi)	0.023	0.19

Legend: MIC—minimal inhibitory concentration, AMC—amoxicillin/clavulanate, * exceptionally ampicillin instead of amoxicillin accepted for early data on amino-penicillin; ** highly resistant strains are non-*P. gingivalis Porphyromonas*; *** complemented by data from Gamboa et al., 2014 [45] applying M.I.C.E on WC, n.d.—not determined, ^$^ result over all antibiotics of this particular class; fluoroquinolones: Cip—ciprofloxacin, Spa—sparfloxacin, Mox—moxifloxacin, Lev—levofloxacin; tetracyclines: Tet—tetracycline, dox—doxycycline; macrolides: Ery—erythromycin, Azi—azithromycin.

**Table 4 antibiotics-10-01438-t004:** Culture conditions used for MIC determination of *Porphyromonas gingivalis*.

Country	Ref.	Year	Method	Agar/Broth	Incubation Time [h]
Finland	[30]	1993	Etest	BBA	96
Japan	[31]	1995	BD	GAB	48–72
Spain	[32]	1998	AD	WC	48
Germany	[33]	1999a	AD	WC	48
Germany	[34]	1999b	Etest	BA	168
International	[35]	2004	AD	WC	48
The Netherlands	[36]	2005a	Etest	BA (Ox no.2)	120
Spain	[36]	2005b	Etest	BA (Ox no.2)	120
Turkey	[37]	2005c	Etest	BBA	48
Brazil	[38]	2006	Etest	BBA	48
Italy	[39]	2007	MD	BB	48
Switzerland	[40]	2008	Etest	BBA	48
Colombia	[41]	2010	Etest	BBA	48–96
Iran	[42]	2011	Etest	BA (Ox no.2)	72–120
The Netherlands	[43]	2012	Etest	BA (Ox no.2)	≥48
Morocco	[44]	2019	Etest	BA (Ox no.2)	72

Legend: Etest—epsilometer agar testing, BD—broth dilution, AD—agar dilution, MD—microdilution; BBA—Brucella blood agar, GAB—Gifu anaerobic broth, WC—Wilkins–Chalgren agar, BA—blood agar (non-selective), Ox—Oxoid, BB—Brucella broth.

## Data Availability

Data is contained within the article and Appendix A.

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
