# Peer review of "The Antimicrobial Susceptibility of Porphyromonas gingivalis: Genetic Repertoire, Global Phenotype, and Review of the Literature"

_antibiotics, 2021, doi:10.3390/antibiotics10121438_

Round 1

Reviewer 1 Report

Dear authors

Thank you very much for your manuscript submission. Indeed, you have done a very interesting work; well-designed and well-performed. However, some items should be revised as below:

1) Your work is an interesting in vitro-in silico study. You have worked on 29 strains of the major periodontal pathogen Porphyromonas gingivalis and three P. gulae. How did you provide these strains? In table 1, the second column refers the Country and Year of isolation. It is confused, because at the same time, only six strains are depicted this study. Moreover, table 1 shows only 24 strains belonging to Porphyromonas gingivalis.

2) It is necessary to show the procedures of this study through a schematic flow chart.  It's absolutely a vital means.

3) You mention TLR4 in different part of manuscript. Please mention the complete term of Toll-like receptor 4 (TLR4) for the first time.

4) As you mention TLR4 in your manuscript it is recommended to read and add the following paper to your manuscript references section.

Toll-Like Receptors: General Molecular and Structural Biology. J Immunol Res. 2021 May 29;2021:9914854. doi: 10.1155/2021/9914854. PMID: 34195298; PMCID: PMC8181103.

Please use the related information regarding TLR4 from this paper within your manuscript.

5) Why did you use  ARG-ANNOT 2014 and CARD 2020 in your study? please explain it very carefully.

6) What is n.d. in tables 2 and 3?

7) Please add a figure relating to E-test.

8) It is recommended to use statistical calculations to have an influential interpretation to support your conclusion.

Author Response

Dear reviewer 1,

thank you very much for taking your time to review our manuscript; please see our point-by-point response in the document attached.

Sincerely, yours

Georg Conrads (in the name of all authors)

Reviewer 2 Report

The study evaluated the in vitro antimicrobial susceptibility of clinical Porphyromonas gingivalis (P. gingivalis) and 3 P. gulae (as an ancestor) to nine antibiotics (amoxicillin, amoxicillin/clavulanate, clindamycin, metronidazole, moxifloxacin, doxycycline, azithromycin, imipenem, and cefoxitin) by E-testing of MIC. The results demonstrated that no antibiotic resistance was observed. Moreover, the authors compared their results with 16 international studies reporting MICs. The results showed that no antibiotic resistance was observed for amoxicillin + clavulanic acid, cefoxitin and imipenem, while some strains were moderately sensitive or resistant to amoxicillin and metronidazole. In addition, the study screened for resistance genes through the published genome of P. gingivalis. The results showed that some resistance genes were detected, such as erm(B) or erm(F) and tet(Q). The authors therefore concluded that antimicrobial resistance of P. gingivalis has not yet emerged. But it should be noted that certain antibiotics show increased MIC and correspond to the detected resistance genes.

Overall, this is a very interesting and comprehensive study. I It compares with other studies and uses databases to find resistance genes. It provides a good summary of the current clinical antibiotic resistance of P. gingivalis.

I have some minor concerns.

  1. Although this study is only compared with the E-testing study using MIC, in the discussion section, it should also be compared with results using other MIC methods. I suggest that the following papers should be cited.

1)Antibiotic Susceptibility Patterns of Aggregatibacter actinomycetemcomitans and Porphyromonas gingivalis Strains from Different Decades. Kulik EM, Thurnheer T, Karygianni L, Walter C, Sculean A, Eick S. Antibiotics (Basel). 2019. PMID: 31817588.

2)Antimicrobial resistance of Aggregatibacter actinomycetemcomitans, Porphyromonas gingivalis and Tannerella forsythia in periodontitis patients. Ardila CM, Bedoya-García JA. J Glob Antimicrob Resist. 2020. PMID: 32169683.

  1. The patient's condition, such as disease status, can affect the strain. The study needs to provide patient information. And the strains sample size is too small.
  2. In Table 3, the references need to be provided.
  3. There are a few grammatical errors throughout the text. And it would be nice to revise carefully to make the text more fluent and avoid any type of grammar or syntactical mistakes.

Author Response

Dear reviewer no. 2,

thank you very much for taking your time to review our manuscript; please see our point-by-point response in the document attached.

Sincerely, yours

Georg Conrads (in the name of all authors)

Reviewer 3 Report

The authors explored the in vitro antimicrobial susceptibility of 29 strains of P. gingivalis and 3 P. gulae to nine antibiotics and reviewed 16 international studies from 1993 until recently. The paper is well organized with in-depth discussion. However, the results can be further improved to increase the novelty and significance of current work.

  1. It has been reported that there is a trend of increasing antibiotic non-susceptibilities in P. gingivalis. How about the trend in this study? What's the statistical significance?
  2. Does the antimicrobial susceptibility of P. gingivalis to different antibiotic change along the time? What's the difference?
  3. Is that possible to explore the antimicrobial resistance genes in the strains used for vitro experiment? Does it echo the analysis of the 77 currently available genomes? Is there any signature genes related with the resistance to specific antibiotic. For example, Azithromycin.

Author Response

Dear Reviewer no. 3,

thank you very much for taking your time to review our manuscript; please see our point-by-point response in the document attached.

Sincerely, yours

Georg Conrads (in the name of all authors)

Round 2

Reviewer 1 Report

Dear Authors
Thank you very much for your rigorous revision. However, there are still some cases which should be revised as below:

  1. I recommend you to add this explanation

    "Most “historical strains” are from the ACTA collection and Dong Mei Deng (a co-author) provided them with agreement of her institution. However, we acknowledge the pioneer work of colleagues: “the authors wish to thank … the many colleagues at ACTA and worldwide for providing strains and/or samples”. Iii) We invested quite some time to identify the most plausible country and year of isolation. That was done by a “Google scholar” search for strain name but also searching e.g. historical strain collection information. For instance on the first view, the famous W83 strain is from the UK as the nation of many publication subjecting it. According to ATCC the origin of W83 is Bonn-Germany. Iv)"

    to Material and Methods section; subtitle "Bacterial strains and antibacterial susceptibility testing".
  2. Please recheck your added flow chart. It seems some represented numbers (n=...) are not correct.
  3. Page 3, lines 56 and 57: ... by blocking Toll-Like Receptor 4 (TLR4) ...
  4. Your explanation regarding ARG-ANNOT 2014 and CARD 2020 is invaluable. However, the mentioned reference for ARG-ANNOT in your manuscript belongs to 2014. For this reason, this is called ARG-ANNOT 2014.
  5. Regarding E-test figure and statistical calculations, I do not persist; I refer it to the editors' decision.

Author Response

Dear Reviewer No. 1,

thank you for improving our manuscript substantially; please find enclosed the point-by-point response. The new manuscript should be available now, with our first line of changes highlighted in yellow and the second line of changes highlighted in blue (addressing your comments here) for overview.

In the name of all authors

Georg Conrads
